# 3D-Printed Polycaprolactone Implants Modified with Bioglass and Zn-Doped Bioglass

**DOI:** 10.3390/ma16031061

**Published:** 2023-01-25

**Authors:** Izabella Rajzer, Anna Kurowska, Jana Frankova, Renáta Sklenářová, Anna Nikodem, Michał Dziadek, Adam Jabłoński, Jarosław Janusz, Piotr Szczygieł, Magdalena Ziąbka

**Affiliations:** 1Department of Mechanical Engineering Fundamentals, Faculty of Mechanical Engineering and Computer Science, University of Bielsko-Biala, 43-300 Bielsko-Biała, Poland; 2Department of Medical Chemistry and Biochemistry, Faculty of Medicine and Dentistry, Palacký University Olomouc, 77515 Olomouc, Czech Republic; 3Department of Mechanics, Materials and Biomedical Engineering, Faculty of Mechanical Engineering, Wroclaw University of Science and Technology, 50-370 Wrocław, Poland; 4Department of Glass Technology and Amorphous Coatings, Faculty of Materials Science and Ceramics, AGH University of Science and Technology, 30-059 Kraków, Poland; 5Faculty of Chemistry, Jagiellonian University, 31-007 Kraków, Poland; 6Department of Ceramics and Refractories, Faculty of Materials Science and Ceramics, AGH University of Science and Technology, 30-059 Kraków, Poland

**Keywords:** bioglass, biomaterials, bone scaffolds, implants, polycaprolactone, 3D-printing, zinc

## Abstract

In this work, composite filaments in the form of sticks and 3D-printed scaffolds were investigated as a future component of an osteochondral implant. The first part of the work focused on the development of a filament modified with bioglass (BG) and Zn-doped BG obtained by injection molding. The main outcome was the manufacture of bioactive, strong, and flexible filament sticks of the required length, diameter, and properties. Then, sticks were used for scaffold production. We investigated the effect of bioglass addition on the samples mechanical and biological properties. The samples were analyzed by scanning electron microscopy, optical microscopy, infrared spectroscopy, and microtomography. The effect of bioglass addition on changes in the SBF mineralization process and cell morphology was evaluated. The presence of a spatial microstructure within the scaffolds affects their mechanical properties by reducing them. The tensile strength of the scaffolds compared to filaments was lower by 58–61%. In vitro mineralization experiments showed that apatite formed on scaffolds modified with BG after 7 days of immersion in SBF. Scaffold with Zn-doped BG showed a retarded apatite formation. Innovative 3D-printing filaments containing bioglasses have been successfully applied to print bioactive scaffolds with the surface suitable for cell attachment and proliferation.

## 1. Introduction

Osteochondral repair involves a combination of cartilage and bone tissue engineering [1]. Consideration of the needs of both tissues is essential in designing successful osteochondral tissue implants [2]. Scaffolds provide a 3D environment that is desirable for the production of cartilage and bone tissues. Ideally, the scaffold (1) should have controlled degradation; (2) allow neovascularization and the diffusion of nutrients, oxygen, and waste products; (3) promote cell viability, differentiation, and ECM production; (4) adhere and integrate with the surrounding native cartilage and bone; and (5) provide the mechanical integrity depending on the defect location [3,4,5]. Nasal fractures are the most common facial injuries and may involve the nasal bones and the cartilaginous structures [6]. Damage or loss of facial cartilage and bone is an important problem for patients and also for the laryngologist and plastic surgeons. Injured natural cartilage is slow and difficult to heal and has almost no ability to regrow itself [7]. The requirements for scaffolding materials use in tissue engineering are well-defined. However, the ability to produce such materials has been limited. The lack of vascular networks that do not hinder efficient nutrient transport and the inherent complexity of the composition of the osteochondral tissue make it difficult to successfully repair this region [2].

Numerous scaffold fabrication techniques have been investigated for cartilage and bone regeneration such as solvent casting, salt leaching, freeze drying, fiber bonding, nonwoven fabrics production, electrospinning, and gas foaming [8,9]. However, these methods have limitations in manual interaction, difficulty in control of complicated internal architecture, and reproducibility as well as toxicity concerns due to using organic solvents. In contrast, fabrication of a tissue scaffold using 3D-printing technology is very promising because these processes allow the fabrication of scaffolds with complex geometries and internal architecture [10].

FDM (fused deposition modeling) is one of the most widely used rapid prototyping systems in the world [2]. The main reasons for its increasing popularity and use have been its reliability, safe and simple fabrication process. Traditional scaffold fabrication methods result in structures of random internal architecture and great variation from part to part. Due to the repeatability of the 3D-printing process, a more thorough investigation into the influence of the internal microarchitecture on cellular responses is available [11].

At present, a number of researchers are working with (FDM), i.e., a material extrusion process. Conventional extrusion process uses granular or pelletized feedstock. Granule based FDM 3D-printers are very expensive and usually constitute the equipment of specialized laboratories. The great advantage of the using granular feedstock is the possibility to mix different materials to create composite scaffolds [12,13]. Most of the work carried out with PCL and 3D-printing requires the use of granulate/powder and is associated with time-consuming preparation of the material [14]. Other researchers, such as Hassanajili et al. are trying to combine 3D-printing and freeze-drying to produce microporous PLA/PCL/HA scaffolds for bone tissue engineering [15]. In the FDM/FFF based extrusion process, generally filaments are used as a standard feedstock material [16]. Feedstock filaments are usually made of amorphous thermoplastics having typical diameter in the range 1.7–2.8 mm. Even though many 3D printers (commercial and otherwise) employ the FDM/FFF approach using commonly available polymers, utilization of 3D printing to fabricate scaffolds using medical grade filaments remains largely unexplored [17]. While the FDM process of a few thermoplastics and their composite materials derived from granular feedstock material has been well demonstrated and explored, there are still several challenges in developing new FDM/FFF PCL composite filaments as feedstock materials which will simplify the production of composite scaffolds for tissue engineering. Usually in commercial printers the printing process is tightly connected with their own supplied materials, which may not be suitable for biomedical applications [18]. Moreover, filament materials are generally supplied in spools which are not entirely used during the production of small implants for cartilage or bone replacement. In order to reduce the wasting of unused filament and to create the possibility of producing complex osteochondral scaffolds the aim of this work was to develop the novel method of producing PCL/BG composite filament sticks. Filament sticks can be joined together and used in commercially available 3D printers for the production of a scaffold that supports the regeneration of the bone tissue region in osteochondral defect.

Polycaprolactone (PCL) has attracted much attention in FDM printing of tissue scaffolds [19]. PCL is a non-toxic polyester that is biocompatible and biodegradable. PCL has a longer degradation time than the other polymers from the group of aliphatic polyesters. Thus, biomaterials made using this polymer can be implanted in areas subjected to increased load. The use of PCL, which has better mechanical parameters, to make scaffolds for the treatment of tissue defects allows sufficient space to be maintained for extracellular matrix formation, not only until the moment of producing new tissue, but until it obtains the required mechanical parameters. The good solubility of PCL, its low melting point (59–64 °C), long-term degradation properties (>24 months to lose total mass), and exceptional blend-compatibility have stimulated extensive research into its potential applications in the biomedical field [20]. PCL is usually selected for its mechanical stability, biocompatibility, and resorbability. However, PCL has limited bioactivity. Therefore, the incorporation of bioactive glasses (BGs) into PCL has been suggested as a wise approach to improve surface hydrophilicity.

BGs are revolutionary biomaterials that show great potential in a wide range of biomedical applications from hard to soft tissues. The classical use of BGs in bone tissue management results from their ability to bond to living bone (bioactivity) and to promote bone regeneration through the release of biologically active ions (osteoinductivity). Moreover, dissolution products of BGs can also induce angiogenesis and enhance cell functions such as cell migration and proliferation, which is particularly desirable for both hard as well as soft tissue regeneration [21,22]. One of the great advantages of BGs is the possibility to incorporate other biologically active ions into their structure which, when released in a biological environment, enhance the therapeutic effect [23]. One of these ions is zinc (Zn^2+^). Zinc is the second most predominant trace element in the human body playing an important role in the regulation of the cell cycle and cell division, while functioning as an intracellular signaling molecule, antioxidant, and co-factor of numerous enzymatic reactions and proteins. Importantly, zinc ions show anti-inflammatory and antibacterial properties [24]. It was shown that the incorporation of Zn^2+^ ions into BG significantly improved the antibacterial effect against Gram-positive and Gram-negative bacteria [25,26]. Recent studies showed that Zn^2+^ ions released from BG exhibit an immunomodulatory capacity by influencing M1/M2 macrophage polarization [27].

The main objective of this study was to develop 3D-printed scaffolds for bone defect treatment as a future component of an osteochondral implant.

Here, to the best of our knowledge, we presented the first investigation aiming at the development of bioactive PCL_BG and PCL_BG_Zn short composite filaments. These filaments will serve as feedstock materials for the commercial 3D printer. Our long-term goal is to fabricate bioactive, complex, osteochondral implants via 3D printing.

Recently, we were successful in developing pure PCL filament sticks using injection molding and successfully used them for FDM [28]. As a further improvement, we produced composite filaments made of a PCL matrix with 0.5, 5, 10 percentages of graphene as a filler [29,30]. We have already proved that the addition of small amounts of graphene, bioglass or zinc-doped bioglass significantly enhance PCL antibacterial efficacy [31] and that the presence of Zn^2+^ ions in the electrospun polycaprolactone membranes influence the osteogenic differentiation of cells [32].

In this work, as a step toward the development of 3D printed scaffolds for bone region in osteochondral tissue regeneration, composite filaments obtained by injection molding for the production of porous scaffolds by low-cost FDM technology were evaluated. The first part of the work focused on the development of a composite filament modified with BG and Zn-doped BG. The success of our work depended on the careful selection of the proportions of polymer materials and bioactive molecules (BG, Zn-doped BG) to result in the desired properties of the composite stick. The main outcome was the manufacture of strong and flexible filament sticks of the required length, diameter, and properties. Then, the composite filament sticks were used for scaffold production. In this study, we investigated the effect of BG and Zn-doped BG addition on the mechanical and biological properties of 3D-printed scaffolds.

## 2. Materials and Methods

### 2.1. Materials

The bioactive glasses in the systems 40SiO_2_–54CaO–6P_2_O_5_ and 49CaO–5ZnO–6P_2_O_5_–40SiO_2_ (mol.%), labeled as BG (A2) and BG_Zn (A2Zn5), respectively, were prepared by the sol-gel method [33,34]. The following reagents were used for syntheses: tetraethyl orthosilicate (TEOS, Si(OC_2_H_5_)_4_), triethyl phosphate (TEP, OP(OC_2_H_5_)_3_) (Sigma-Aldrich, Merck Life Science Sp.z.o.o., Poznań, Poland), calcium nitrate tetrahydrate (Ca(NO_3_)_2_·4H_2_O), zinc nitrate hexahydrate (Zn(NO_3_)_2_·6H_2_O), and 1 M HCl solution (POCh, Gliwice, Poland). The resulting sols were left at ambient conditions to undergo gelation. After that, the as-synthesized gels were dried at 80 °C and calcinated at 700 °C. BG particles of size 1.5 µm (d50) were obtained by milling in an attritor equipped with ZrO_2_ grinding balls in isopropyl alcohol medium. Particle size distribution was investigated using Mastersizer 2000 equipment (Malvern, UK).

PCL (Mn 80 kDa) in granular form was purchased from Merck (Warszawa, Poland). The polymer granules were dry-mixed together with BG and BG_Zn powders in order to obtain 0.4 wt% of BG addition in the mixture. Mechanical stirring of the blend for 20 min was applied. PCL_BG and PCL_BG_Zn blends were used for the injection molding process.

### 2.2. Filament Fabrication

Injection molding was carried out on a Babyplast 6/10P (Rambaldi, Molteno, Italy) machine. A series of injection molding experiments were performed to evaluate the influence of the processing parameters on the quality of the polymer sticks. Furthermore, injection molding simulation using SolidWorks plastic software (29.3.0.0059, Dassault Systemes, Paris, France) was performed [28]. The injection molding parameters are presented in Table 1.

A mold made of stainless steel was used during the tests. Figure 1a shows the CAD model of the mold used in the research. The mold produced 12 elements in the form of sticks (filaments) within one cycle (Figure 1b,d). After cooling the injection mold, the part was removed and the filament in the form of sticks was separated from it. Properly designed stick ends would allow them to be combined into one filament that can be used in traditional 3D printers (Figure 1c).

### 2.3. Scaffold Printing

The 3D models of scaffolds were designed using Autodesk Inventor Professional 2016 software (2020, Autodesk, Inc., San Rafael, CA, USA) and exported to a.stl file (compatible with any 3D printer software). In the Ultimaker Cura 3.6.0 software (3.6, Ultimaker, Utrecht, The Netherlands), the models were prepared for printing by selecting the 3D printer type (Anet A8), defining the printing parameters, dividing the model into layers, and saving the settings in a G-code file, which is a set of commands to be read and executed by the 3D printer.

The nozzle temperature during printing for particular filaments was 170–190 °C. The temperature of the print bed was 50–20 °C, depending on the filament. In the case of the BG-modified filament, the print bed temperature for the first layer was much higher to improve the adhesion. The layer thickness of 0.2 mm and the printing speed of 3.75–7.5 mm/s were set. A detailed list of printing parameters for particular biomaterials is presented in Table 2.

### 2.4. Analysis and Testing

Microscopic observations of the produced sticks and scaffolds were carried out using an Opta-Tech (Warszawa, Poland) optical microscope and an Opta-Tech stereomicroscope (Warszawa, Poland), equipped with a CMOS 3 camera and OptaView 7 software.

The microstructure of the samples before and after incubation in simulated body fluid (SBF) was also observed using scanning electron microscopy (SEM). The ultra-high-resolution scanning microscope Nova NanoSEM 200 (FEI Europe Company, Eindhoven, Netherlands) with the Genesis XM X-ray microanalysis system (EDAX, Tilburg, the Netherlands) featuring the EDAX Sapphire Si(Li) EDS detector was used. The samples were stuck onto a conductive carbon tape and coated with a 10-nm carbon layer (EMACE600 sputter coater, Leica Microsystems, Wetzlar, Germany). The observations took place in low vacuum conditions, using the low vacuum secondary electron detector with an accelerated voltage of 10–18 kV.

The chemical structure of the sticks was analyzed by FTIR spectroscopy (FTIR Bio-Rad FTS60V spectrophotometer Bio-Rad, Warszawa, Poland) using ATR mode in the range of 4000–600 cm^−1^ with a resolution of 4 cm^−1^.

The distribution of BG and BG_Zn particles inside the PCL sticks and scaffolds was examined using high-resolution X-ray tomography (µCT). The tests were carried out on the 1172 SkyScan, Bruker^®^. Each sample was recorded with a resolution of 5.5 µm (lamp parameters: 34 kV, 210 µA). The quantitative analysis of selected parameters was performed in the CTAn program. The computed tomography system, by making a series of sections along the perpendicular axis of the sample, collected data that were used to reconstruct the image.

Mechanical testing was performed by using a Hegewald und Peschke Inspekt Table Blue 5 kN machine (Hegewald und Peschke, Nossen, Germany). Tensile tests were carried out according to ISO 7500-1 with a cross-head speed of 5 mm/min. For each test, a minimum of six samples were used. During testing, tensile strength, elastic modulus, and maximum strain at break were determined.

### 2.5. Scaffold In Vitro Degradation

To evaluate the bioactivity of the printed samples, the scaffolds were soaked in SBF solution at 37 °C for up to 14 days. The SBF was refreshed every two days to maintain its composition. The SBF solution was prepared according to Kokubo [35] with a 1.5× standard ion composition. 1.5× SBF concentration was used to accelerate the biomineralization processes. After 7 and 14 days, samples were removed from the SBF solution and stored in a desiccator prior to SEM, EDS, and µCT analyses.

### 2.6. Cell Culture Study

The human osteoblastic osteosarcoma cell line SaOS-2 was cultivated in McCoy’s medium supplemented with 10% FBS and antibiotics (1% penicillin and streptomycin) at 37 °C in 5% CO_2_ atmosphere. The medium was changed every 48 h. After reaching 80% confluence, the cells were washed with sterile PBS, released by incubation with 0.25% trypsin solution, and centrifuged for 5 min at 1300 rpm and room temperature. Then, the cells were resuspended in 10 mL of culture medium, counted, and seeded on appropriate materials [36]. For the experiments, the cells were used in 2.-10. passage. For the 24-h incubation period, the cells were cultured in serum-free medium.

#### 2.6.1. Preparation of Materials

The cell culture experiment was carried out with three types of printed scaffolds: (1) PCL, (2) PCL_BG, and (3) PCL_BG_Zn. The selected materials were sterilized by soaking in 70% ethanol for 30 min and by exposure to UV light for 20 min (each side) and then washed with sterile water.

#### 2.6.2. Cell Viability

The sterile membranes were placed at the bottom of 96-well culture plates and seeded with cells (SaOS-2) at a concentration of 1 × 10^5^ cells/mL for 24-h cultivation and 1 × 10^4^ cells/mL for the 7-day culture period. Cell viability (24 h and 7 days) was assessed using an MTT assay, which determined the mitochondrial reduction of MTT (3-(4,5-dimethyltiazol-2 yl) 2,5 diphenyltetrazolium bromide) to formazan. The absorbance was measured at 540 nm.

#### 2.6.3. Cell Morphology and the Biocompatibility of Tested Materials

Biocompatibility and cell morphology for the tested materials were studied using fluorescence (Olympus BX40 microscope, Olympus, Tokyo, Japan) and confocal microscopy (Carl Zeiss LSM780 Spectral Confocal, Carl Zeiss AG, Oberkochen, Germany). Cell viability was evaluated by acridine orange after 1 and 14 days of cell culture using fluorescence microscopy. The cells were stained for 1 min with 0.01% acridine orange solution, rinsed with PBS, and photographed.

The morphology of the cells was determined after 1 day of cell culture using confocal microscopy: Alexa Fluor^®^ 555 Phalloidin (Abcam, Cambridge, UK) and DAPI staining. SaOS-2 cultivated on membranes were fixed for 15 min with 4% paraformaldehyde, permeabilized for 10 min with 0.1% Triton X-100 in PBS, and then blocked for 20 min in 3% BSA in PBS. Alexa Fluor^®^ 555 Phalloidin (diluted 1:20 in PBS, Abcam, Cambridge, UK) was applied for 15 min, followed by rinsing with PBS and application of SlowFade mounting medium with DAPI.

All the data were expressed as means ± standard deviation (SD). Statistical analyses were performed by the one-way analysis of variance (one-way ANOVA). The statistical difference was considered statistically significant at *p* < 0.05. Statistically significant differences were indicated by lowercase letters.

## 3. Results

An image of the filament model and the model of the injection mold are shown in Figure 1a–c. The influence of the injection parameters on the final geometry of filament sticks was evaluated using macro- and microscopic observation. As shown in Figure 1d, the shape of the mold was properly reproduced, and composite filament sticks were successfully obtained. The results of microscopic observations are presented in Figure 2. The surface and the cross-section of the pure PCL filament stick were smooth and flat (Figure 2a,e). An even distribution of the BG powder on the surface of the polymer sticks as well as on its cross-section was observed (Figure 2b,f). However, when high magnification was used, small BG particle agglomerates were detected on the PCL_BG composite filament stick surface (Figure 2d). In the case of PCL_BG_Zn filament, the presence of numerous agglomerates was observed (Figure 2c,g).

The distribution of BG and BG_Zn particles inside PCL sticks was examined using high-resolution X-ray tomography (µCT). The µCT test results (Figure 2h–j) confirmed the incorporation of BG and BG_Zn particles into the polymer matrix. In the case of PCL_BG samples, the BG was evenly distributed; however, small agglomerates of the powder were observed (Figure 2i). The presence of numerous clusters of BG with zinc in the PCL_BG_Zn filament was demonstrated (Figure 2j). It can be seen that BG_Zn aggregates of different sizes were present, and these aggregates were not uniformly distributed throughout the PCL_BG_Zn composite stick.

SEM micrographs of the PCL, PCL_BG, and PCL_BG_Zn sticks are shown in Figure 3. The SEM evaluation confirmed the presence of powder agglomerates in the modified sticks (Figure 3b,c), whereas the unmodified PCL filaments were characterized by a smooth surface (Figure 3a). EDX analysis of the PCL_BG filament showed the presence of elements such as C, O, Si, and Ca in PCL_BG samples (Figure 3e) and C, O, Zn, Si, P, and Ca corresponding to the BG-doped with zinc in the PCL_BG_Zn samples (Figure 3g). The microscopic observations, µCT, SEM, and EDX results confirmed the incorporation of BG into the polymer filament sticks.

The ATR-FTIR spectra of filament sticks in the range of 4000–600 cm^−1^ are presented in Figure 4. The characteristic bands for BG and BG doped with Zn (BG_Zn) are shown on their spectra. The band located at 1020 cm^−1^ corresponds to the stretching vibration of [PO_4_] as well as [SiO_4_] units. The presence of weak bands at 930 cm^−1^ corresponds to Si-O stretching of non-bridging oxygen atoms in SiO_4_ tetrahedra. Additionally, the small bands at 600 cm^−1^ and 875 cm^−1^ correspond to the O-P-O and CO_3_^2−^ bending vibrations. The characteristic bands for PCL are observed on the spectra of the composite filament samples. The two peaks at 2949 and 2865 cm^−1^ correspond to asymmetric CH_2_ stretching and symmetric CH_2_ stretching, respectively. The pure PCL filament stick showed the characteristic peaks at 1727 cm^−1^ (carbonyl stretching), 1240 cm^−1^ (asymmetric C-O-C stretching), 1175 cm^−1^ (symmetric C-O-C stretching), 1293 cm^−1^ (C-O and C-C stretching), and 1157 cm^−1^ (C-O and C-C stretching) [32]. The bands characteristic for the BG and BG_Zn powder were not observed on the spectra for the modified polymer sticks. This was probably related to the small amount of BG used to modify the polymer matrix. However, the observation of the microstructure with the use of an optical microscope, high-resolution X-ray tomography (µCT), and scanning electron microscope together with the EDX analysis confirmed the presence of elements indicating the incorporation of the BG into the polymer filament matrix.

The results of mechanical tests of filament sticks are shown in Table 3. The mean values of Young’s modulus, tensile strength, and strain at break for all types of filaments are presented. The mechanical properties of composite sticks strongly depended on the filler dispersion. Young’s modulus of the PCL_BG_Zn and PCL_BG samples were 35% and 11% lower than Young’s modulus of pure PCL polymer sticks. The addition of BG into the sticks increased the strain at the break of the samples. All samples had similar tensile strengths.

Three types of composite filaments (Ø1.75 mm) in the form of sticks were used to produce 3D scaffolds (PCL, PLC_BG, PCL_BG_Zn). Sticks were joined together and applied in a commercially available 3D printer. The printed scaffold consisted of three levels. Each level was made of parallel bars with a square cross-section (side length—1 mm) spaced 0.7 mm apart. The bars on adjacent levels were perpendicular to each other, resulting in cube-shaped pores. The virtual model is shown in Figure 5a. The volume, porosity, and total surface area of the obtained structure were calculated in reference to a solid cuboid with the same external dimensions. The results are presented in Figure 5b.

Macro and micro images of the obtained scaffolds are presented in Figure 5c–f. Macroscopic observations showed that the obtained scaffolds were consistent with the previously designed virtual models. All scaffolds had a smooth surface and an open, uniform, and interconnected porous structure (Figure 5c). Microscopic analysis revealed the presence of additives in the printed scaffolds (Figure 5e,f). On the surface of the PCL_BG_Zn sample, clusters of Zn-doped BG particles were observed.

The mean values of tensile strength, Young’s modulus, and strain at the break of the scaffolds are presented in Table 4. The highest Young’s modulus was observed for the PCL_BG_Zn sample (about 140 MPa). The highest strain at break was observed for the PCL_BG sample. The tensile strength of the scaffolds compared to the strength of filaments was lower by 59%, 58%, and 61%, respectively, for PCL, PCL_BG, and PCL_BG_Zn samples. The mechanical tests showed that the printing process caused a decrease in Young’s modulus and tensile strength for all samples. The presence of a porous, spatial microstructure affected the mechanical properties by reducing them.

The bioactivity of the printed scaffolds was evaluated in in vitro tests by immersion in SBF. Figure 6 shows micrographs of the printed PCL, PCL_BG, and PCL_BG_Zn scaffolds. From Figure 6a, it can be seen that the PCL scaffold after 7 days of immersion in SBF fluid was characterized by a rough surface, but no apatite formation was observed (Figure 6d). The surface of BG-modified scaffolds (Figure 6b) was covered by a uniform and dense Ca-P layer composed of globules. Scaffolds containing Zn-doped BG showed less apatite formation on the surface (Figure 6c) compared to PCL_BG. The EDX analysis confirmed the presence of Ca–P minerals within the PCL_BG and PCL_BG_Zn scaffolds (Figure 6e,f).

In order to evaluate alterations in the 3D microstructure of scaffolds after incubation in SBF for 14 days, a high-resolution µCT was used. Three-dimensional images of the scaffolds before and after immersion in SBF were obtained, and the microstructural parameters were determined. The µCT method allowed for the analysis of the internal geometry of the scaffolds, including the size and the dispersion of bioactive additives. The µCT images of the selected PCL_BG_Zn scaffolds before and after incubation in SBF and the results of the analysis are presented in Figure 7 and Figure 8, and Table 5.

In the sample PCL_BG_Zn before incubation (Figure 7), 1596 inclusions were detected, which was 0.4% of the scaffold volume. The average thickness of inclusions was 46 µm, and about 20% of inclusions were in the thickness range of 13–22 µm.

In the PCL_BG_Zn sample, after 14 days of incubation in SBF, 3697 incubations were detected (Figure 8), which was 0.7% of the scaffold volume. The average thickness of inclusions was 39 µm, and about 24% of inclusions were in the thickness range of 13–22 µm.

A greater number of inclusions in the sample after 14 incubations favored the formation of an apatite, which was confirmed by SEM-EDS observations and analysis.

The viability of the cells seeded on printed tested materials was evaluated by MTT assay (Figure 9). As seen from the graph, the results demonstrated no significant difference between the cells cultivated on the tested materials.

The in vitro results demonstrated the interaction between the cells and tested materials. After the incubation period, the cells were adhered to and spread out on the surface of the samples. Acridine orange staining indicated the viable cells after the 24-h and 14-day incubation periods (green color). Alexa Fluor^®^ 555 Phalloidin and DAPI fluorescent staining (after 24 h) showed that SaOS-2 cells were well adhered and spread out on the surface of the samples. The cells were evenly distributed and maintained their morphology (red color of cytoskeleton, Figure 10).

## 4. Discussion

The possibility of using composite sticks to produce a 3D scaffold for bone region regeneration of osteochondral tissue was investigated. The PCL sticks were modified with BG and Zn-doped BG. Zn^2+^ ions play a significant role in the formation, development, mineralization, and maintenance of healthy bones [37]. It was expected that Zn incorporation would improve the osteogenic ability of PCL scaffold. The composite filament sticks were successfully obtained using injection molding. The microscopic observations (optical microscope, stereomicroscope, and SEM) confirmed that BG and Zn-doped BG were successfully incorporated into the filament. This was supported by µCT images of the composite sticks. Although the particles agglomerated, creating local extremes, globally they were well distributed in the polymer matrix. However, the deterioration of mechanical properties was observed. The agglomeration of particles may be generated due to nozzle clogging and the limited mixing capacity in the injection molding machine. In order to improve the properties, the mixing time of the blend before the injection process will be extended in the future. However mechanical properties of all scaffold studies were within the range of application in bone tissue. One important feature of our innovative sticks is that they can be joined together and applied in a standard FDM printer. We have successfully used the obtained filaments and produced composite 3D scaffolds with a smooth surface and an open, uniform, and interconnected porous structure. In bone tissue engineering, the pore interconnection of the scaffold plays an important role in bone ingrowth because it conducts cells and vessels between pores [38]. In the produced scaffolds, the interconnectivity between the pores was adequate to facilitate angiogenesis and promote good vascularization, fixation, proliferation, and cell differentiation [39]. The bioactivity assay originally proposed by Kokubo is one of the most commonly used tests to indirectly evaluate the biocompatibility of BGs. The SBF solution simulated the human blood plasma inducing the mineralization of an apatite layer on the surface of the bioactive materials [35]. The results of bioactivity evaluation demonstrated that the incorporation of BG into PCL sticks played an important role in the nucleation and growth of apatite on the surface of 3D-printed scaffolds. However, the scaffold with Zn-doped bioactive glass showed a retarded apatite formation. Our previous work also revealed that modification of BG with zinc ions delayed the formation of apatite, especially at the early stage of incubation in SBF [32]. This could be explained by the fact that in SBF, Zn^2+^ ions prevent apatite nucleation by binding to active growth sites of apatite [37]. The µCT analysis revealed that the number of inclusions in the PCL_BG_Zn samples increased after 14 days of incubation in SBF fluid, which confirmed the formation of a calcium phosphate layer on the sample surface. The analysis of the scaffold geometry confirmed the spatial and porous structure. The potential of the PCL_BG materials for osteochondral repair was determined by investigating their influence on osteoblast biocompatibility in vitro. The cells adhered well, showed osteoblast morphology, and expanded onto the printed scaffolds. Based on the MTT analysis, it was shown that the materials were not toxic to the cells, and after 1 day and 14 days, all tested scaffolds had viable cells in the samples (green color). However, there were no significant differences between them.

### Future Research Directions

The ability to connect different filament sticks during the printing process enables the production of a variety of scaffolds without having to change the spool, as is the case with the traditional FDM process. In the future, it will be possible to mix the polymer sticks with different additives (also with drugs and other antibacterial and bioactive particles) to create complex, graded scaffolds with a mechanical and biological gradient of properties for osteochondral defect treatment. The development and increasing availability of modern technologies and their transfer to medical applications as well as the possibility of achieving high results using commonly available equipment makes treatment rooms look for new (better and cheaper) solutions. Therefore, it is likely that in the coming years operating rooms will be equipped with 3D printers, which (thanks to the solution proposed in this work) will allow 3D printing of the patient specific implants.

## 5. Conclusions

Innovative 3D-printed filaments containing BG and Zn-doped BG have been successfully produced and can be used in the future for osteochondral tissue regeneration. We have obtained strong and flexible filament sticks of the required length, diameter, and properties. The filaments were used for the 3D printing of bioactive composite scaffolds. The BG and BG_Zn particles were successfully incorporated into the printed scaffolds and were visible on their surface. Test performed in SBF proved the bioactivity of the composite scaffolds. However, the mineralization process on Zn-doped BG was retarded. The µCT analysis confirmed the formation of a calcium phosphate layer on the PCL_BG_Zn scaffold after 14 days. Initial cell culture studies confirmed that the surface of scaffolds was suitable for cell attachment and proliferation.

The FDM 3D-printing method has great potential in the field of regenerative medicine for the fabrication of defect-filling scaffolds for tissue regeneration. One of the advantages of FDM is that a wide range of biodegradable and biocompatible filament materials can be printed. However, the filaments are produced in a spool with one type of modifier. Our idea is to assemble differently modified filament sticks to produce more complex implants. The positive results of the preliminary investigation will allow for the development of more complex scaffold systems in the future.

## 6. Patents

Application number: P.428429; Filing date: 31 December 2018. Patent number: Pat.240243. Date of granting the right: 12 September 2021. Authors: Izabella Rajzer, Adam Jabłoński, Anna Kurowska, Jerzy Kopeć, Marcin Sidzina. “A method of producing implants from bioresorbable thermoplastic polymer composites, especially in the form of 3D scaffolds, intended for the reconstruction of cartilage and bone tissue defects, using 3D printing”.

## Figures and Tables

**Figure 1 materials-16-01061-f001:**
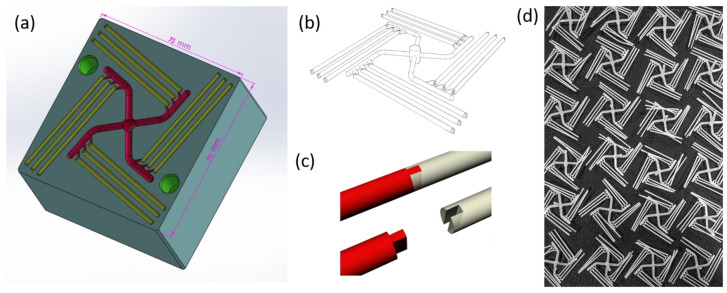
(**a**) CAD model of the mold used in the research, (**b**) designed model of the part, (**c**) scheme of connecting sticks, (**d**) produced parts with sticks.

**Figure 2 materials-16-01061-f002:**
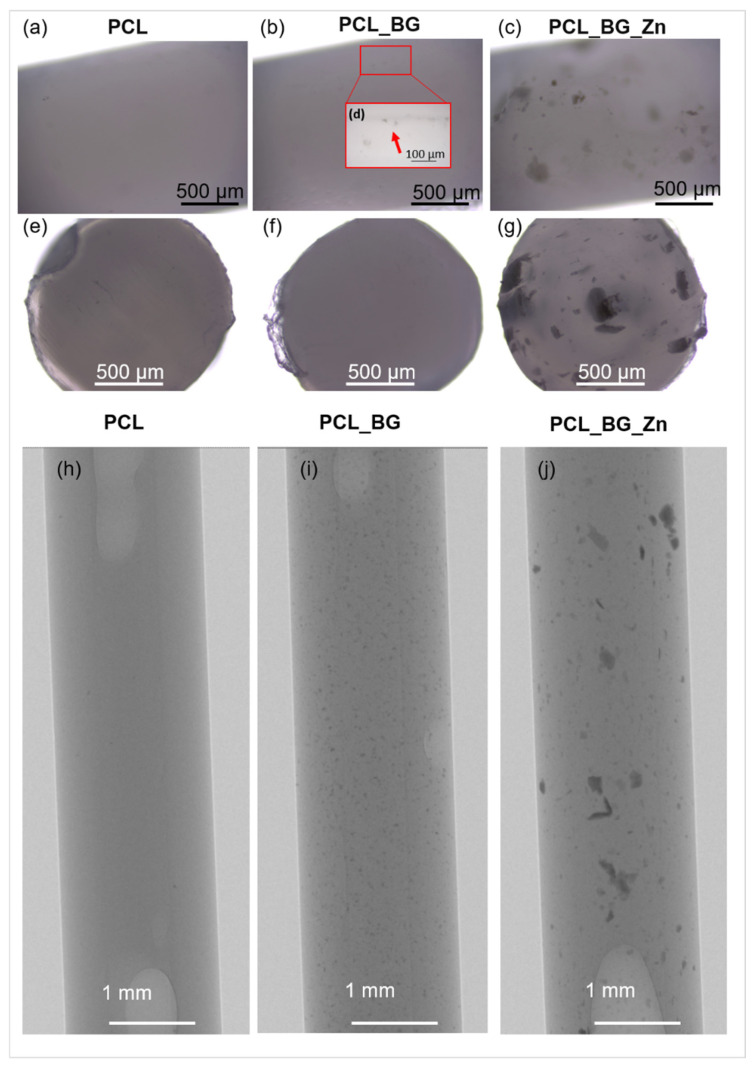
Microscopic images of obtained filaments and their cross-sections: (**a**,**e**) PCL; (**b**,**d**,**f**) PCL_BG; (**c**,**g**) PCL_BG_Zn. µCT images of (**h**) PCL, (**i**) PCL_BG, (**j**) PCL_BG_Zn sticks.

**Figure 3 materials-16-01061-f003:**
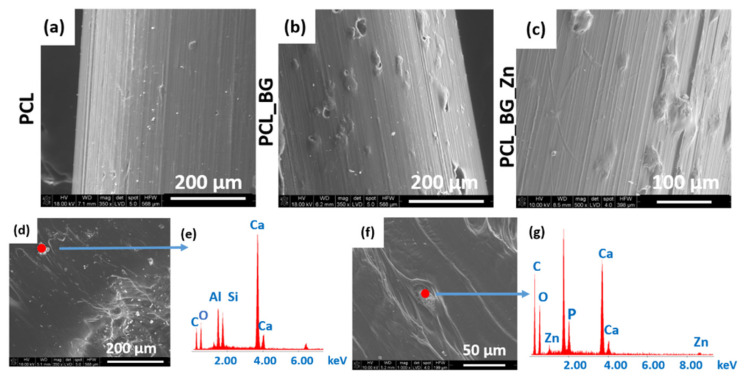
SEM micrographs of (**a**) PCL; (**b**) PCL_BG; (**c**) PCL_BG_Zn filaments along with the results of EDX analysis (**d**,**e**) PCL_BG; (**f**,**g**) PCL_BG_Zn.

**Figure 4 materials-16-01061-f004:**
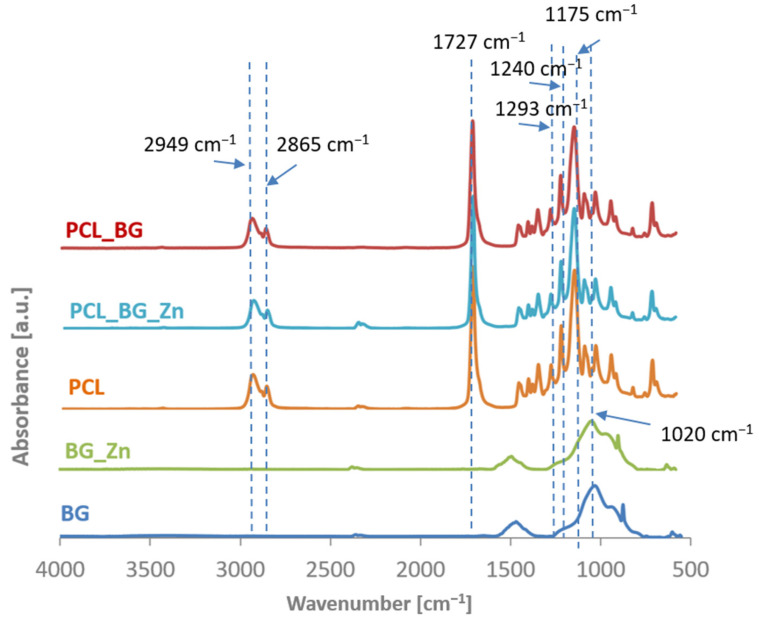
ATR-FTIR spectra of PCL_BG; PCL_BG_Zn; PCL; filaments, BG_Zn powder, and BG powder.

**Figure 5 materials-16-01061-f005:**
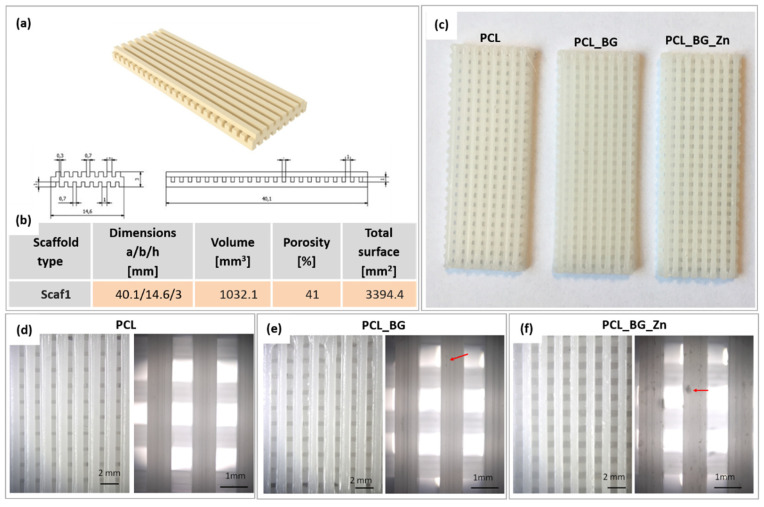
(**a**) Scaffold model; (**b**) the topographical scaffold properties; (**c**) scaffolds after the printing process—macroscopic view; (**d**,**e**,**f**) microscopic images of PCL, PCL_BG, PCL_BG_Zn scaffolds.

**Figure 6 materials-16-01061-f006:**
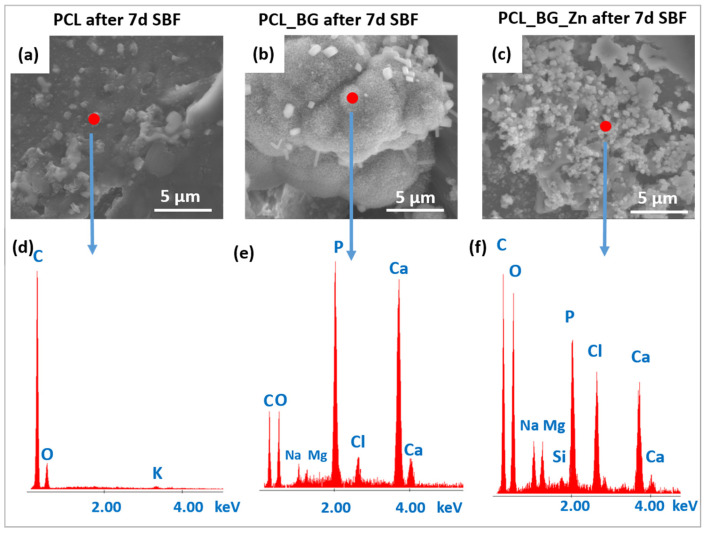
SEM micrographs of (**a**) PCL; (**b**) PCL_BG; (**c**) PCL_BG_Zn scaffolds after 7 days of incubation in SBF. EDX analysis of the aforementioned samples (**d**,**e**,**f**).

**Figure 7 materials-16-01061-f007:**
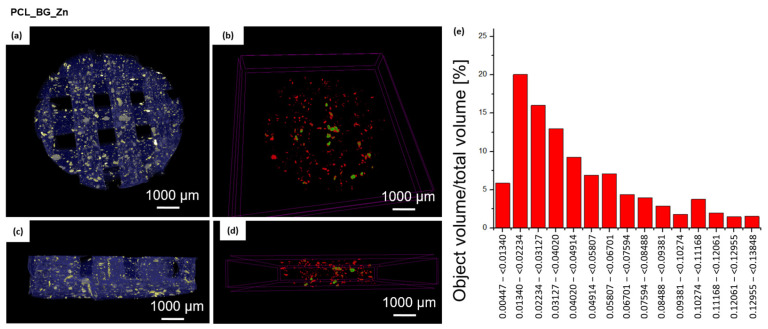
µCT images of sample PCL_BG_Zn before incubation in SBF. Images showing the top view (**a**) and the cross-section of the scaffold (**c**). The scaffold volume is marked in blue; the distribution of the modifying particles is highlighted in yellow. Distribution of BG_Zn particles without the polymer matrix (**b**,**d**) (particles marked in red and green). Histogram of the distribution of particles (**e**).

**Figure 8 materials-16-01061-f008:**
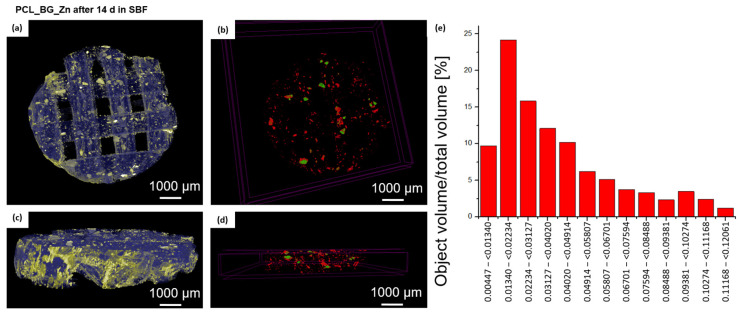
µCT images of sample PCL_BG_Zn after incubation in SBF. Images showing the top view (**a**) and the cross-section of the scaffold (**c**). The scaffold volume is marked in blue; the distribution of the modifying particles is highlighted in yellow. Distribution of BG_Zn particles without the polymer matrix (**b**,**d**) (particles marked in red and green). Histogram of the distribution of particles (**e**).

**Figure 9 materials-16-01061-f009:**
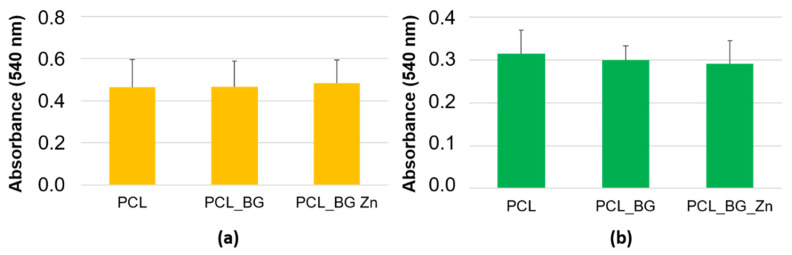
Cell viability was assessed by MTT assay. The results show a comparison of the cell viability of SaOS-2 cultured on individual materials for 24 h (**a**) and 7 days (**b**). Number of measurements: n = 9.

**Figure 10 materials-16-01061-f010:**
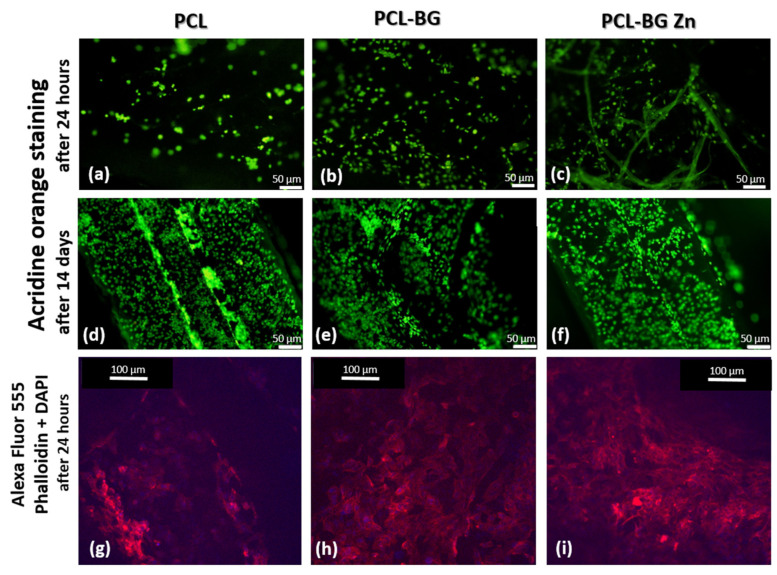
Acridine orange (green color indicates the viable cells) staining, (**a**) cells cultivated 24 hours on PCL samples, (**b**) cells cultivated 24 hours on PCL-BG, (**c**) cells cultivated 24 hours on PCL-BG Zn, (**d**) cells cultivated 14 days on PCL samples, (**e**) cells cultivated 14 days on PCL-BG, (**f**) cells cultivated 14 days on PCL-BG Zn Alexa Fluor^®^ 555 Phalloidin (red color indicates the cytoskeleton), and DAPI (blue color indicates the nucleus) staining (**g**) cells cultivated 24 hours on PCL samples, (**h**) cells cultivated 24 hours on PCL-BG samples, (**i**) cells cultivated 24 hours on PCL-BG Zn samples.

**Table 1 materials-16-01061-t001:** Parameters of the injection process of PCL and PCL_BG and PCL_BG_Zn blends.

Parameters	PCL	PCL_BG	PCL_BG_Zn
Injection size [mm]	17.5	17.1	23
Cooling time [s]	60	60	55
First injection pressure [bar]	130	130	100
Time of the first injection pressure [s]	3.3	4.5	4.5
Second injection pressure [bar]	120	90	90
Time of the second injection pressure [s]	8	4	4
Plastification temperature [°C]	210	230	200
Chamber temperature [°C]	205	205	190
Nozzle temperature [°C]	190	190	180

**Table 2 materials-16-01061-t002:** Printing process parameters for PCL, PCL_BG, and PCL_BG_Zn filaments.

	Nozzle Temperature [°C]	Print Bed Temperature—First Layer [°C]	Print Bed Temperature—Next Layers [°C]	Printing Speed [mm/s]
PCL	170	40	40	7.5
PCL_BG	170	50	20	3.75
PCL_BG_Zn	190	50	20	7.5

**Table 3 materials-16-01061-t003:** Mechanical properties of PCL stick filaments: PCL, PCL_BG, PCL_BG_Zn.

Sample	Young Modulus [mpa]	Tensile Strength [mpa]	Strain at Break [%]
PCL	289 ± 45	16.30 ± 0.79	114.23 ± 73.10
PCL_BG	257 ± 88	15.24 ± 1.34	155.62 ± 121.69
PCL_BG_Zn	188 ± 56	16.49 ± 1.02	143.85 ± 40.76

**Table 4 materials-16-01061-t004:** Mechanical properties of scaffolds made of PCL stick filaments: PCL, PCL_BG, PCL_BG_Zn.

Sample	Young’s Modulus [MPa]	Tensile Strength [MPa]	Strain at Break [%]
PCL	136 ± 5	6.70 ± 0.33	29.92 ± 26.53
PCL_BG	130 ± 4	6.37 ± 0.39	81.16 ± 55.06
PCL_BG_Zn	140.5 ± 9	6.35 ± 0.39	32.25 ± 27.36

**Table 5 materials-16-01061-t005:** Calculation results based on µCT images.

Sample	Object Volume/Total Volume (Obj.V/TV) [%] *	Object Surface/Volume Ratio [mm^2^/mm^3^]	Structure Thickness St. Th [mm]	Number of Objects Obj. N
PCL_BG_Zn	0.436	121.631	0.046	1596
PCL_BG_Zn_ (14dSBF)	0.747	146.715	0.039	3697

* This parameter allowed us to determine which volume of the entire scaffold was occupied by the polymer and reinforcement particles [%].

## Data Availability

Data are contained within the article.

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
