# Peer review of "3D-Printed Polycaprolactone Implants Modified with Bioglass and Zn-Doped Bioglass"

_materials, 2023, doi:10.3390/ma16031061_

Round 1

Reviewer 1 Report

1.      The abstract requires the addition of quantitative results.

2.      Given the “take-home” message at the end of the abstract, the present form was insufficient.

3.      Put the keywords in a new order based on alphabetical order.

4.      Nothing truly unique in its current state. Because of the lack of novel, the current study looks to be a replication or modified study. The authors must describe their novel in detail. This work should be rejected owing to a major issue.

5.      It is essential to summarize previous studies' merits, novelties, and limitations in the introductory part to emphasize the gaps in the research that the latest research seeks to address.

6.      Why the p[resent study performs experimental testing? Why not combined with Computational Simulation and Analytical Formula? It needs more explanation. Also, further study adopting computational simulation needs to be included that offer several advantages such as lower cost and faster results. It is a vital topic that authors must provide in the introduction and/or discussion section. Additionally, the MDPI's suggested reverence should be taken to substantiate this explanation as follows: Ammarullah, M. I.; Santoso, G.; Sugiharto, S.; Supriyono, T.; Wibowo, D. B.; Kurdi, O.; Tauviqirrahman, M.; Jamari, J. Minimizing Risk of Failure from Ceramic-on-Ceramic Total Hip Prosthesis by Selecting Ceramic Materials Based on Tresca Stress. Sustainability 2022, 14, 13413. https://doi.org/10.3390/su142013413

7.      To enhance the understandability of the section on materials and methods easier for them to understand rather than just depending on the main text as it exists at the moment, the authors could add additional illustrations in the form of figures that explain the workflow of the present study.

8.      Other information about the tool, such as the manufacturer, country, and specifications, should be provided.

9.      Error and tolerance of experimental tools used in this work are important information that needs to be explained in the manuscript. It is would use as a valuable discussion due to different results in the further study by other researcher.

10.   A comparative assessment with similar previous research is required.

11.   The discussion in present article is extremely poor in quality as overall. The authors must elaborate on their arguments and provide a thorough justification. Don't just state the results and give a quick explanation.

12.   Please include the limitation of the present study, it is missing.

13.   The reference needs to be enriched from the literature published five years back. MDPI reference is strongly recommended.

14.   The manuscript needs to be proofread by the authors since it has grammatical and language issues.

15.   The graphical abstract should be provided in the system after modification of peer review.

Author Response

Bielsko-Biala,12.12.2022

Corresponding Author:

Izabella Rajzer

University of Bielsko-Biala

Willowa 2, 43-309 Bielsko-Biala, Poland

e-mail: irajzer@ath.bielsko.pl

On behalf of my co-authors, we thank you very much for giving us an opportunity to revise our manuscript. We appreciate Reviewers very much for their comments and suggestions on our manuscript. Those comments are all valuable and very helpful for revising and improving our paper. We have made revision which marked in red in the paper. Attached please find the revised version, which we would like to submit for your kind consideration.

Reviewer 1

Comments and Suggestions for Authors

  1. The abstract requires the addition of quantitative results.

As suggested by the Reviewer, the quantitative results were added to the abstract.

  1. Given the “take-home” message at the end of the abstract, the present form was insufficient.

“take-home” message was added at the end of the abstract as follows:
Innovative 3D printing filaments containing bioglasses has been successfully applied to print bioactive scaffolds with the surface suitable for cell attachment and proliferation.

  1. Put the keywords in a new order based on alphabetical order.

As suggested by the Reviewer, the keywords were placed in alphabetical order.

It was: 3D printing, biomaterials, implants, bone scaffolds, polycaprolactone, bioglass, zinc.

It is: bioglass, biomaterials, bone scaffolds, implants, polycaprolactone, 3D-printing, zinc.

  1. Nothing truly unique in its current state. Because of the lack of novel, the current study looks to be a replication or modified study. The authors must describe their novel in detail. This work should be rejected owing to a major issue.

It is essential to summarize previous studies' merits, novelties, and limitations in the introductory part to emphasize the gaps in the research that the latest research seeks to address.

As suggested by the Reviewer, an analysis of previous studies was performed in order to highlight the novelty presented in this paper:

At present, a number of researchers are working with (FDM). i.e.: a material extrusion process. Conventional extrusion process uses granular or pelletized feedstock. Granule based FDM 3D-printers are very expensive and usually constitute the equipment of specialized laboratories. The great advantage of the using granular feedstock is the possibility to mix different materials to create composite scaffolds. Liu at al. prepared a PCL/TCP/PEG scaffold through 3D-printing to build a hydrophilic surface and improve the osteogenic activity [12]. The 3D-printed scaffolds were prepared using a 3D Bioprinting system, where the powdery printing materials were injected into the printer barrel, heated to set temperature, and extruded through a nozzle. Before that, biomaterials were mixed with dichloromethane, stirred for 3 h, poured into the mold, and dried at room temperature. The dry mixed materials were frozen and smashed to obtain the powdery printing materials [12]. Wang at al. used FDM 3D printer to print PCL scaffolds modified with bioglass [13]. In order to obtain different contents of bioactive glass/PCL composite materials, different amounts of bioactive glass were blended with PCL and extruded through the nozzle. In another study, Wang at al. used a viscous solution of PCL dissolved in acetone as the carrier phase to disperse the bioactive glass microparticles to formulate a homogeneous ink. Prepared material was fed in the direct ink writing (DIW), and the printed struts solidified rapidly in the cold ethanol due to the solubility change of PCL, thus facilitating the fabrication of the PCL/Cu-BaG composite scaffolds [14]. Most of the work done with PCL and 3D printing requires the use of granulate/powder and is associated with time-consuming preparation of the material. Other researchers, such as Hassanajili et al. are trying to combine 3D-printing and freeze-drying to produce microporous PLA/PCL/HA scaffolds for bone tissue engineering [15]. In the FDM/FFF based extrusion process, generally filaments are used as a standard feedstock material [16]. Feedstock filaments are usually made of amorphous thermoplastics having typical diameter in the range 1.7–2.8 mm. Even though many 3D-printers (commercial and otherwise) employ the FDM/FFF approach using commonly available polymers, utilization of 3D-printing to fabricate scaffolds using medical grade filaments remains largely unexplored [17]. While FDM process of few thermoplastics and their composite materials derived from granular feedstock material has been well demonstrated and explored, there are still several challenges in developing new FDM/FFF PCL composite filaments as feedstock materials which will simplify the production of composite scaffolds for tissue engineering. Usually in commercial printers the printing process is tightly connected with their own supplied materials, which may not be suitable for biomedical applications [18]. Moreover, filament materials are generally supplied in spools which is not entirely used during the production of small implants for cartilage or bone replacement. In order to reduce the wasting of not used filament and to create the possibility of producing complex osteochondral scaffolds the aim of this work was to developed the novel method of producing polycaprolac-tone/bioglass composite filament sticks. Filament sticks can be joint together and used in commercially available 3D-printers for the production of a scaffold that supports the regeneration of the bone tissue region in osteochondral defect.

  1. Why the present study performs experimental testing? Why not combined with Computational Simulation and Analytical Formula? It needs more explanation. Also, further study adopting computational simulation needs to be included that offer several advantages such as lower cost and faster results. It is a vital topic that authors must provide in the introduction and/or discussion section. Additionally, the MDPI's suggested reverence should be taken to substantiate this explanation as follows: Ammarullah, M. I.; Santoso, G.; Sugiharto, S.; Supriyono, T.; Wibowo, D. B.; Kurdi, O.; Tauviqirrahman, M.; Jamari, J. Minimizing Risk of Failure from Ceramic-on-Ceramic Total Hip Prosthesis by Selecting Ceramic Materials Based on Tresca Stress. Sustainability 2022, 14, 13413. https://doi.org/10.3390/su142013413

In our previous study we have created polycaprolactone material data from the literature and use Solidworks Plastic software to simulate the process of injection molding in order to evaluate the influence of operational conditions and mold design on the efficiency of an injection process used to produce PCL filament stick for further 3D printing of facial implants. Thermal and rheological data from the literature were imported to the material database of Solidworks Plastics software for simulating the injection molding process of filament sticks. The influence of several injection molding parameters including melt temperature, injection time, packing time, and packing pressure on the geometry of final part (polymer sticks that will be used for 3D printing) was investigated. These studies were supported by simulation, however, the addition of bioglass causes too many unknowns in the injection symulation process. The result of our work are implants and it would be impossible to produce them without conducting experimental tests.

  1. To enhance the understandability of the section on materials and methods easier for them to understand rather than just depending on the main text as it exists at the moment, the authors could add additional illustrations in the form of figures that explain the workflow of the present study.

The section on materials and methods were written in accordance with the guidelines for the journal and present classical research methods for biomedical materials. The authors see no need to add a figure in this section.

  1. Other information about the tool, such as the manufacturer, country, and specifications, should be provided.

Other information about country and manufacturer were provided.

  1. Error and tolerance of experimental tools used in this work are important information that needs to be explained in the manuscript. It is would use as a valuable discussion due to different results in the further study by other researcher.

The following explanation was added to the manuscript:

All the data were expressed as means ± standard deviation (SD). Statistical analyses were performed by the one-way analysis of variance (one-way ANOVA). The statistical difference was considered statistically significant at p < 0.05. Statistically significant differences were indicated by lowercase letters.

  1. A comparative assessment with similar previous research is required.

A comparative assessment with similar research was performed:

The main objective of this study was to develop the 3D printed scaffolds for bone defect treatment as a future component of an osteochondral implant. Here, to the best of our knowledge, we present the first investigation aiming at the development of bioactive PCL_BG and PCL_BG_Zn short composite filaments. These filaments will serve as feedstock materials for the commercial 3D printer. Our long-term goal is to fabricate bioactive, complex, osteochondral implants via 3D print-ing.

Recently, we were successful in developing a pure PCL filament sticks using in-jection molding and successfully used them for FDM [28]. As a further improvement, we produced composite filaments made of a PCL matrix with 0.5, 5, 10 percentages of graphene as a filler [29, 30]. We have already proved that the addition of small amounts of graphene, bioglass or zinc-doped bioglass significantly enhance PCL anti-bacterial efficacy [31] and that the presence of Zn2+ ions in the electrospun polycapro-lactone membranes influence the osteogenic differentiation of cells [32].

In this work, as a step toward the development of 3D printed scaffolds for bone region in osteochondral tissue regeneration, composite filaments obtained by injection molding for the production of porous scaffolds by low-cost FDM technology were evaluated.

  1. The discussion in present article is extremely poor in quality as overall. The authors must elaborate on their arguments and provide a thorough justification. Don't just state the results and give a quick explanation.

Discussion part was corrected.

  1. Please include the limitation of the present study, it is missing.

The development and increas-ing availability of modern technologies and their transfer to medical applications as well as the possibility of achieving high results using commonly available equipment makes treatment rooms look for new (better and cheaper) solutions. Therefore, it is likely that in the coming years operating rooms will be equipped with 3D-printers, which (thanks to the solution proposed in this project) will allow 3D-printing of the patient specific implants. Unfortunately, medical offices are not yet equipped with 3D printers.

  1. The reference needs to be enriched from the literature published five years back. MDPI reference is strongly recommended.

The references were enriched.

  1. Liu, K.; Sun, J.; Zhu, Q.; Jin, X.; et al. Microstructures and properties of polycaprolactone/tricalcium phosphate scaffolds containing polyethylene glycol fabricated by 3D printing. Ceram Int 2022, 48, 24032–24043. DOI: 10.1016/j.ceramint.2022.05.081.
  2. Wang, C.; Meng, C.; Zhang, Z.; Zhu, Q. 3D printing of polycaprolactone/bioactive glass composite scaffolds for in situ bone repair. Ceram Int 2022, 48, 7491–7499. DOI: 10.1016/j.ceramint.2021.11.293.
  3. Wang, X.; Zhang Molino, B.; Pitkänen, S.; et al. 3D Scaffolds of Polycaprolactone/Copper-Doped Bioactive Glass: Ar-chitecture Engineering with Additive Manufacturing and Cellular Assessments in a Coculture of Bone Marrow Stem Cells and Endothelial Cells. ACS Biomater Sci Eng 2019, 5, 4496−4510. DOI: 10.1021/acsbiomaterials.9b00105.
  4. Hassanajili, S.; Karami-Pour, A.; Oryan, A.; Talaei-Khozani, T. Preparation and characterization of PLA/PCL/HA com-posite scaffolds using indirect 3D printing for bone tissue engineering. Mater Sci Eng C 2019, 104, 109960. DOI: 10.1016/j.msec.2019.109960.
  5. Smirnov, A.; Seleznev, A.; Peretyagin, P.; Bentseva, E.; Pristinskiy, Y.; Kuznetsova, E.; Grigoriev, S. Rheological Char-acterization and Printability of Polylactide (PLA)-Alumina (Al2O3) Filaments for Fused Deposition Modeling (FDM). Materials 2022, 15, 8399. DOI: 10.3390/ma15238399
  6. Ravi, P.; Shiakolas, P.S.; Welch, T.R. Poly-L-lactic acid: pellets to fiber to fused filament fabricated scaffolds, and scaffold weight loss study. Addit Manuf 2017, 16, 167-76. DOI: 10.1016/j.addma.2017.06.002.
  7. Dávila, J.L.; Freitas, M.S.; Inforçatti Neto, P.; Silveira, Z.C.; Silva, J.V.L.; D'Ávila, M.A. Fabrication of PCL/β-TCP scaffolds by 3D mini-screw extrusion printing. J Appl Polym Sci 2016, 133, 1-9. DOI: 10.1002/app.43031.

  1. The manuscript needs to be proofread by the authors since it has grammatical and language issues.

The manuscript was proofread and corrected.

  1. The graphical abstract should be provided in the system after modification of peer review.

The graphical abstract will be provided in the system.

Reviewer 2 Report

Comments on materials-2064692

The manuscript entitled “3D printed polycaprolactone implants modified with bioglass and Zn-doped bioglass.” investigated the composite filament in the form of sticks and 3D printed scaffolds as a future component of an osteochondral implant. The first part of the work was focused on the development of a composite filament modified with bioglass and Zn-doped bioglass. The main outcome was the manufacture of bioactive, strong and flexible filament sticks of the required length, diameter, and properties. Then composite filament sticks were used for scaffold production. In addition to that, the effect of bioglass and Zn-doped bioglass on the mechanical and biological properties of 3D-printed scaffolds was also investigated.

The manuscript is well-written, however, there are several amendments required to be resolved before possibly accepting it for publication which are disclosed below:

·       The introduction has some flaws, and a more detailed novelty of their work should be clearly addressed. Thorough references should be cited in the Introduction, which is missing.

·       It is suggested to reorganize figure 3, please try to avoid leaving empty spaces within the figure.

·       It is recommended to label all the peaks of FTIR spectra (figure 4). Also, please in the figure caption write the sample names in chronological order.

·       The authors should provide more discussions on the mechanisms for performance strategies, which would be beneficial for readers to understand their significance.

·       The authors need to provide a comparison of their work with other reported ones to assure the novelty of their work with proper references.

·       The authors mentioned that the agglomeration was observed, however, but they didn’t discuss what will be the effect of it on the final properties. And how it can be further improved.

·       There is a lot of similar work has been published (Acta Bioeng Biomech. 2021;23(2):131-138; ACS Biomater. Sci. Eng. 2019, 5, 9, 4496–4510; Journal of Materials Science: Materials in Medicine, 30, 80 (2019)), the authors need to emphasis on the novelty of their work and make a strong comparison which is missing in their work.

·       In line 416; “Although particles agglomerated, they were well distributed in polymer matrix”, what do the authors mean to say? The sentence seems confusing. Please explain further with proper references.

·       It is also recommended to add some more references from recent years of the related work which is missing in their reference list. Some references are important to understand the progress of polymer-based composite materials and their advantages: Composites Part A, 2022, 153 (2022) 106734; Composites Part B 176 (2019) 107190.

·       The conclusions are not informative. They do not contain numerical data and must be carefully rewritten. Moreover, the authors are recommended to add a few sentences about the prospective applications of their proposed work.

·       The title of the manuscript should be modified since several articles have quite similar titles.

·       The manuscript contains a few misprints, lost intervals and should be corrected with this respect. Similarly, the language expression in the text needs to be carefully checked and revised. There are some grammatical mistakes.

Author Response

Bielsko-Biala,12.12.2022

Corresponding Author:

Izabella Rajzer

University of Bielsko-Biala

Willowa 2, 43-309 Bielsko-Biala, Poland

e-mail: irajzer@ath.bielsko.pl

On behalf of my co-authors, we thank you very much for giving us an opportunity to revise our manuscript. We appreciate Reviewers very much for their comments and suggestions on our manuscript. Those comments are all valuable and very helpful for revising and improving our paper. We have made revision which marked in red in the paper. Attached please find the revised version, which we would like to submit for your kind consideration.

REVIEWER 2:

The manuscript entitled “3D printed polycaprolactone implants modified with bioglass and Zn-doped bioglass.” investigated the composite filament in the form of sticks and 3D printed scaffolds as a future component of an osteochondral implant. The first part of the work was focused on the development of a composite filament modified with bioglass and Zn-doped bioglass. The main outcome was the manufacture of bioactive, strong and flexible filament sticks of the required length, diameter, and properties. Then composite filament sticks were used for scaffold production. In addition to that, the effect of bioglass and Zn-doped bioglass on the mechanical and biological properties of 3D-printed scaffolds was also investigated.

The manuscript is well-written, however, there are several amendments required to be resolved before possibly accepting it for publication which are disclosed below:

  1. The introduction has some flaws, and a more detailed novelty of their work should be clearly addressed. Thorough references should be cited in the Introduction, which is missing.

As suggested by the Reviewer, the introduction was revised and references were made to studies by other researchers.

At present, a number of researchers are working with (FDM). i.e.: a material ex-trusion process. Conventional extrusion process uses granular or pelletized feedstock.  Granule based FDM 3D printers are very expensive and usually constitute the equipment of specialized laboratories. The great advantage of the using granular feedstock is the possibility to mix different materials to create composite scaffolds. Liu at al. pre-pared a PCL/TCP/PEG scaffold through 3D printing to build a hydrophilic surface and improve the osteogenic activity [12]. The 3D-printed scaffolds were prepared using a 3D Bioprinting system, where the powdery printing materials were injected into the printer barrel, heated to set temperature, and extruded through a nozzle. Before that, biomaterials were mixed with dichloromethane, stirred for 3 h, poured into the mold, and dried at room temperature. The dry mixed materials were frozen and smashed to obtain the powdery printing materials [12]. Wang at al. used FDM 3D printer to print PCL scaffolds modified with bioglass [13]. In order to obtain different contents of bioactive glass/PCL composite materials, different amounts of bioactive glass were blended with PCL and extruded through the nozzle. In another study, Wang at al. used a viscous solution of PCL dissolved in acetone as the carrier phase to disperse the bioactive glass microparticles to formulate a homogeneous ink. Prepared material was fed in the direct ink writing (DIW), and the printed struts solidified rapidly in the cold ethanol due to the solubility change of PCL, thus facilitating the fabrication of the PCL/Cu-BaG composite scaffolds [14]. Most of the work done with PCL and 3D printing requires the use of granulate/powder and is associated with time-consuming preparation of the material. Other researchers, such as Hassanajili et al. are trying to combine 3D printing and freeze-drying to produce microporous PLA/PCL/HA scaffolds for bone tissue engineering [15]. In the FDM/FFF based extrusion process, generally filaments are used as a standard feedstock material. Feedstock filaments are usually made of amorphous thermoplastics having typical diameter in the range 1.7–2.8 mm. Even though many 3D printers (commercial and otherwise) employ the FDM/FFF approach using commonly available polymers, utilization of 3D printing to fabricate scaffolds using medical grade filaments remains largely unexplored [16]. While FDM process of few thermoplastics and their composite materials derived from granular feedstock material has been well demonstrated and explored, there are still several challenges in developing new FDM/FFF PCL composite filaments as feedstock materials which will simplify the production of composite scaffolds for tissue engineering. Usually in commercial printers the printing process is tightly connected with their own supplied materials, which may not be suitable for biomedical applications [17]. Moreover, filament materials are generally supplied in spools which is not entirely used during the production of small implants for cartilage or bone replacement. In order to reduce the wasting of not used filament and to create the possibility of producing complex osteochondral scaffolds the aim of this work was to developed the novel method of producing polycaprolac-tone/bioglass composite filament sticks. Filament sticks can be joint together and used in commercially available 3D printers for the production of a scaffold that supports the regeneration of the bone tissue region in osteochondral defect.

  1. Liu, K.; Sun, J.; Zhu, Q.; Jin, X.; et al. Microstructures and properties of polycaprolactone/tricalcium phosphate scaffolds containing polyethylene glycol fabricated by 3D printing. Ceram Int 2022, 48, 24032–24043. DOI: 10.1016/j.ceramint.2022.05.081.
  2. Wang, C.; Meng, C.; Zhang, Z.; Zhu, Q. 3D printing of polycaprolactone/bioactive glass composite scaffolds for in situ bone repair. Ceram Int 2022, 48, 7491–7499. DOI: 10.1016/j.ceramint.2021.11.293.
  3. Wang, X.; Zhang Molino, B.; Pitkänen, S.; et al. 3D Scaffolds of Polycaprolactone/Copper-Doped Bioactive Glass: Ar-chitecture Engineering with Additive Manufacturing and Cellular Assessments in a Coculture of Bone Marrow Stem Cells and Endothelial Cells. ACS Biomater Sci Eng 2019, 5, 4496−4510. DOI: 10.1021/acsbiomaterials.9b00105.
  4. Hassanajili, S.; Karami-Pour, A.; Oryan, A.; Talaei-Khozani, T. Preparation and characterization of PLA/PCL/HA com-posite scaffolds using indirect 3D printing for bone tissue engineering. Mater Sci Eng C 2019, 104, 109960. DOI: 10.1016/j.msec.2019.109960.
  5. Smirnov, A.; Seleznev, A.; Peretyagin, P.; Bentseva, E.; Pristinskiy, Y.; Kuznetsova, E.; Grigoriev, S. Rheological Char-acterization and Printability of Polylactide (PLA)-Alumina (Al2O3) Filaments for Fused Deposition Modeling (FDM). Materials 2022, 15, 8399. DOI: 10.3390/ma15238399
  6. Ravi, P.; Shiakolas, P.S.; Welch, T.R. Poly-L-lactic acid: pellets to fiber to fused filament fabricated scaffolds, and scaffold weight loss study. Addit Manuf 2017, 16, 167-76. DOI: 10.1016/j.addma.2017.06.002.
  7. Dávila, J.L.; Freitas, M.S.; Inforçatti Neto, P.; Silveira, Z.C.; Silva, J.V.L.; D'Ávila, M.A. Fabrication of PCL/β-TCP scaffolds by 3D mini-screw extrusion printing. J Appl Polym Sci 2016, 133, 1-9. DOI: 10.1002/app.43031.
  8. It is suggested to reorganize figure 3, please try to avoid leaving empty spaces within the figure.

Figure 3 was reorganized.

It was:

Figure 3. SEM micrographs of (a) PCL; (b, c) PCL_BG; (e, f) PCL_BG_Zn filaments together with EDX analysis (d, g).

It is:

Figure 3. SEM micrographs of (a) PCL; (b) PCL_BG; (c) PCL_BG_Zn filaments together with EDX analysis (d, e) PCL_BG; (f, g) PCL_BG_Zn.

  1. It is recommended to label all the peaks of FTIR spectra (figure 4). Also, please in the figure caption write the sample names in chronological order.

The peaks of FTIR spectra were labeled and the samples names were put in chronological order.

Figure 4. ATR-FTIR spectra of PCL; PCL_BG; PCL_BG_Zn filaments, bioglass powder (BG) and bioglass doped with Zn (BG_Zn powder).

Figure 4. ATR-FTIR spectra of PCL_BG; PCL_BG_Zn; PCL filaments, bioglass doped with Zn (BG_Zn) and bioglass (BG).

  1. The authors should provide more discussions on the mechanisms for performance strategies, which would be beneficial for readers to understand their significance.

More discussion was provided.

  1. The authors need to provide a comparison of their work with other reported ones to assure the novelty of their work with proper references.

A comparision with other works was added to the manuscript.

  1. The authors mentioned that the agglomeration was observed, however, but they didn’t discuss what will be the effect of it on the final properties. And how it can be further improved.

Althought particles aglomerated, creating local extremes, globally they were well distributed in polymer matrix. The deterioration of mechanical properties was observed. This agglomeration may be generated due to nozzle clogging and the limited mixing capacity in the injection molding machine. In order to improve the properties, we can extend the mixing time of the blend before the injection process. However mechanical properties of all scaffolds studies are within the range of application in bone tissue. Hance the fabricated scaffolds have good mechanical properties for application in bone tissue engineering.

  1. There is a lot of similar work has been published (Acta Bioeng Biomech. 2021;23(2):131-138; ACS Biomater. Sci. Eng. 2019, 5, 9, 4496–4510; Journal of Materials Science: Materials in Medicine, 30, 80 (2019)), the authors need to emphasis on the novelty of their work and make a strong comparison which is missing in their work.

We have added following explanation to the main manuscript:

In our previous work, a filament made of a PCL was developed using injection molding and successfully used for FDM [Materials 2022, 15(20), 7295]. As a further improvement, we produced composite filament made of a PCL matrix with 0.5, 5, 10 percentages of graphene as a filler [Journal of Materials Science, 2020, 55(9), pp. 4030–4042; International Journal of Molecular Sciences, 2022, 23(18), 10899]. In this work, as a step toward the development of 3D printed scaffolds for bone region in osteochondral tissue regeneration, composite filaments modified with bioglass and Zn-doped bioglass for the production of porous scaffolds by low-cost FDM technology were evaluated. We have already proved that the addition of small amounts of bioglass, zinc-doped bioglass or graphene significantly enhance antibacterial efficacy [Acta Bioeng Biomech. 2021;23(2):131-138] and that the presence of Zn2+ in the electrospun polycaprolactone membranes influence the osteogenic differentiation of cells [Journal of Materials Science: Materials in Medicine, 30, 80 (2019)].

However no physicochemical and mechanical characterization of the PCL_BG and PCL_BG_Zn filament and scaffolds were done and no studies on the effect of bioglasses addition on mineralization process and cell morphology of the 3D printed scaffolds have been performed.

In this work, as a step toward the development of 3D printed scaffolds for bone region in osteochondral tissue regeneration, composite filaments modified with bioglass and Zn-doped bioglass for the production of porous scaffolds by low-cost FDM technology were evaluated.

  1. In line 416; “Although particles agglomerated, they were well distributed in polymer matrix”, what do the authors mean to say? The sentence seems confusing. Please explain further with proper references.

As observed in µ-CT analyses, scaffolds with BG_Zn addition have higher agglomeration. But as we mensioned above (point 6) globally they were well distributed in polymer matrix.

  1. It is also recommended to add some more references from recent years of the related work which is missing in their reference list. Some references are important to understand the progress of polymer-based composite materials and their advantages: Composites Part A, 2022, 153 (2022) 106734; Composites Part B 176 (2019) 107190.

We have added more references form recent years. References mentioned by Reviewer are not connected with the subject of article. We read the mentioned articles and we think that it's not ethical to suggest posting them:

  1. Uddin, D. Estevez, F.X. Qin, From functional units to material design: A review on recent advancement of programmable microwire metacomposites, Composites Part A: Applied Science and Manufacturing, Volume 153, 2022, 106734, https://doi.org/10.1016/j.compositesa.2021.106734.

This review presents a low-profile strategy based on designing a single wire functional unit and assembling it in a specific arrangement to develop a range of microwire composites with programmable EM responses.

  1. Uddin, F.X. Qin, D. Estevez, S.D. Jiang, L.V. Panina, H.X. Peng, Microwave programmable response of Co-based microwire polymer composites through wire microstructure and arrangement optimization, Composites Part B: Engineering, Volume 176, 2019, 107190, https://doi.org/10.1016/j.compositesb.2019.107190.

This work presents a facile strategy based on a single component tunable medium to program the transmission response over wide frequency bands. Structural modification of one type of microwire through suitable current annealing and arrangement of the annealed wires in multiple combinations were sufficient to distinctly red-shift the transmission dip frequency of the composites.

  1. The conclusions are not informative. They do not contain numerical data and must be carefully rewritten. Moreover, the authors are recommended to add a few sentences about the prospective applications of their proposed work.

We have added prospective application of our work.

  1. The title of the manuscript should be modified since several articles have quite similar titles.

We have checked the title of the manuscript and none of the articles found had the same title. We decided to leave the title as it is.

  1. The manuscript contains a few misprints, lost intervals and should be corrected with this respect. Similarly, the language expression in the text needs to be carefully checked and revised. There are some grammatical mistakes.

The manuscript was carefully check and revised.

Reviewer 3 Report

Authors fabricated PCL implants modified with BG and Zinc doped BG through 3D printing technology for the application of osteochondral implant is interesting. The composite materials were characterized by SEM and FTIR. The incorporation of BG and Zinc doped BG to the PCL scaffold was not clearly proven by FTIR. EDS results alone is not enough to confirm the presence of inorganic elements. Additional characterization methods like TGA and DSC can be used to confirm the presence of inorganic materials present in the scaffold quantitatively.

1)     Authors should provide the physical composition of PCL/BG and PCL/BG-Zn clearly.

2)     Please give more explanation for the decrease in mechanical properties upon addition of bioglass for filaments but for scaffolds was almost increased.

3)     All heading styles should be uniform as per journal style.

4)     Table style should be match with journal format.

5)     Scale bar should be provided in Figures 7 and 8

6)     In Figure 9 legend, “with each other for 24 hours (a) and 7 days (b)”. Please check whether it is 7 or 14 days. Please check throughout the manuscript.

Author Response

Bielsko-Biala,12.12.2022

Corresponding Author:

Izabella Rajzer

University of Bielsko-Biala

Willowa 2, 43-309 Bielsko-Biala, Poland

e-mail: irajzer@ath.bielsko.pl

On behalf of my co-authors, we thank you very much for giving us an opportunity to revise our manuscript. We appreciate Reviewers very much for their comments and suggestions on our manuscript. Those comments are all valuable and very helpful for revising and improving our paper. We have made revision which marked in red in the paper. Attached please find the revised version, which we would like to submit for your kind consideration.

 Authors fabricated PCL implants modified with BG and Zinc doped BG through 3D printing technology for the application of osteochondral implant is interesting. The composite materials were characterized by SEM and FTIR. The incorporation of BG and Zinc doped BG to the PCL scaffold was not clearly proven by FTIR. EDS results alone is not enough to confirm the presence of inorganic elements. Additional characterization methods like TGA and DSC can be used to confirm the presence of inorganic materials present in the scaffold quantitatively.

  • Authors should provide the physical composition of PCL/BG and PCL/BG-Zn clearly.

Chemical composition was provided in the material section:

Bioglass: 40SiO2–54CaO–6P2O5 (mol.%)

Zn doped bioglass: 49CaO–5ZnO–6P2O5–40SiO2 (mol.%)

  • Please give more explanation for the decrease in mechanical properties upon addition of bioglass for filaments but for scaffolds was almost increased.

It cannot be unequivocally confirmed that the decrease in mechanical properties upon addition of bioglass for filament and increased for scaffolds are statistically significant because the results are within the limits of the measurement error. However, mechanical properties of all scaffolds studies are within the range of application in bone tissue. Hance the fabricated scaffolds have good mechanical properties for application in bone tissue engineering.

  • All heading styles should be uniform as per journal style.

Heading styles were corrected.

4)     Table style should be match with journal format.

Table style was corrected.

5)     Scale bar should be provided in Figures 7 and 8

Scale bar was added.

6) In Figure 9 legend, “with each other for 24 hours (a) and 7 days (b)”. Please check whether it is 7 or 14 days. Please check throughout the manuscript.

We have check and corrected

Round 2

Reviewer 1 Report

Reviewers greatly appreciate the efforts that have been made by the author to improve the quality of their articles after peer review. I reread the author's manuscript and further reviewed the changes made along with the responses from previous reviewers' comments. Unfortunately, the authors failed to make some of the substantial improvements they should have made making this article not of decent quality with biased, not cutting-edge updates on the research topic outlined. In addition, the author also failed to address the previous reviewer's comments, especially on comments number 4 (nothing really novel with significant contribution in additive manufacturing field in cutting-edge insight) 5 (not a comprehensive discussion, suggested literature not incorporated), and 10 (comprehensive discussion related to results meaning and impact in depth should be discussed). With all due respect, the reviewer opposed this article to be published and must be rejected. Thank you very much for the opportunity to read the author's current work.

Author Response

This article has been revised in accordance with the valuable comments of 3 reviewers. The authors do not consider the article mentioned by Reviewer 1 entitled: „Minimizing Risk of Failure from Ceramic-on-Ceramic Total Hip Prosthesis by Selecting Ceramic Materials Based on Tresca Stress” to be extremely relevant to this article. Moreover we have added other MDPI references to make the introduction more understandable. We emphasized the novely of our research, according to Reviewer suggestion. So far, no one has produced a filament using the injection method that allows to obtain variously modified implants, including bioglass and bioglass with zinc (we have been granted a patent and in order to obtain it, innovative research had to be carried out). Computer simulation is essential prior to fabrication process, and here the implants are susessfully produced. We are disappointed that the Reviewer does not consider our achievements important to the readers of the Materials.

Reviewer 2 Report

The reviewer has no more queries to ask. The authors have answered all the queries. The manuscript can be accepted for publication.

Author Response

The authors thank the reviewer for help in improving the article.

Reviewer 3 Report

The Manuscript was revised well according to the reviewer comments. I feel the introduction part is too big and it is getting bored to the readers. I suggest that the introduction part need to be concised. 

Author Response

The Authors thank the Reviewer for help in improving the manuscript. The introduction has been extended at the requests of other reviewers.